# Unresolved CFD and DEM Coupled Solver for Particle-Laden Flow and Its Application to Single Particle Settlement

**Seongjin Song** [1] and **Sunho Park** [1,2,*]

1   Department of Convergence Study of Ocean Science and Technology, Korea Maritime and Ocean University, Busan 49112, Korea; ssong@g.kmou.ac.kr
2   Department of Ocean Engineering, Korea Maritime and Ocean University, Busan 49112, Korea
*   Correspondence: spark@kmou.ac.kr; Tel.: +82-51-410-4329

**Abstract:** In the present study, a single particle settlement was studied using a developed unresolved computational fluid dynamics (CFD) and discrete element method (DEM) coupling solver. The solver was implemented by coupling OpenFOAM, the open-source computational fluid dynamics libraries, with LIGGGHTS, the open-source discrete element method libraries. An averaging method using a kernel function was considered to decrease the grid dependency. For the drag model of a single particle, a revised volume fraction with a particle volume expansion coefficient was applied. Falling particles in a water tank were simulated and compared with the empirical correlation. A parametric study using several integrated added mass coefficients and volume expansion coefficients from low to high Reynolds numbers was carried out. The simulations which used the developed numerical methods showed significantly improved predictions of particle settlement.

**Keywords:** particle-laden flows; Euler-Lagrange methods; computational fluid dynamics; discrete element method; particle settling velocity; OpenFOAM; LIGGGHTS

## 1. Introduction

Particle-laden flows have been studied for several decades due to the wide range of related industrial applications. Approaches to a two-phase fluid modeling for the particle-laden flow can be categorized into the Euler-Euler approach and the Euler-Lagrangian approach [1]. The Euler-Euler approach was based on an interpenetrating continuum for the fluid and particles, while the Euler-Lagrangian approach tracked a particle using the motion equations of the particle [2]. The Euler-Lagrangian approach is the one most commonly applied in various engineering fields to model particle-laden flows [3–5].

Computational fluid dynamics (CFD) and discrete element method (DEM) coupling solvers that use the Euler-Lagrangian approach have been popular. Cundall and Strack [6] presented a basic concept of DEM, which simplified the collision between particles using a spring-dashpot model. The simple spring dashpot model could handle a large number of particle collisions [7,8]. To understand the interaction between fluid and particle, DEM has been coupled with CFD methods such as the finite volume (FVM), finite difference (FDM), lattice Boltzmann (LBM), and meshless methods [9–12]. The computational efficiency and convenience of CFD and DEM coupled solvers are advantageous over other coupled solvers [1].

CFD and DEM coupled solvers can be classified into following resolved or unresolved approaches. In resolved CFD and DEM coupled solvers, the fluid force acting on a particle can be calculated integrating the pressure and velocity fields in its surface. For that purpose, a dense grid is needed to

obtain accurate fluid flow in CFD [13–15]. As such, its applicability to particle-laden flow with a higher number of particles has been limited due to the large computational cost [16].

The unresolved approach of the CFD and DEM coupled solvers used an empirical drag model based on the relative velocity and volume fraction of the fluid flow. Particle tracking was updated using an averaging method in each cell [17]. A dense grid was not needed to obtain an accurate drag force [7]. The unresolved solver showed a high level of computational efficiency for bulk particle-laden flows [3,7]. For very dilute flows, the dependence of the volume fraction in the drag model could be negligible when cell size was larger than the particles [5]. The particle diameter needed to be at least one-third smaller than the cell size in CFD [18]. In non-uniform cells and several particle sizes, the unresolved solvers caused issues in terms of accuracy and reliability [3,19,20]. Many researchers have studied numerical methods to overcome this grid dependency [21–24].

A single particle settling was one of the standard benchmark problems for particle-laden flows. A single particle was released under gravity. Once released, the particle falling speed increased from zero and finally reached a constant settling velocity. Many simulations selected the single settling problem as the benchmark problem because of simple physics. Feng and Michaelides [25] and Ten Cate et al. [26] used the Lattice Boltzmann Method (LBM) coupled with the immersed boundary method, and Alapti et al. [27] used the smoothed profiled method, to predict the settling velocity. Recently, unresolved CFD and DEM coupling methods have been used [5,22,28–30].

The purpose of this paper is to study the fluid and particle interaction using the above-mentioned benchmark problem. To study the interaction, the unresolved CFD and DEM coupled solver was developed using open-source computational fluid dynamics libraries, OpenFOAM version 5.x, and open-source discrete element method libraries, LIGGGHTS [31]. LIGGGHTS stands for LAMMPS Improved for General Granular and Granular Heat Transfer Simulations. The Gaussian kernel function-based averaging method was applied to decrease the grid dependency of the results obtained. The revised volume fraction method was presented in the drag force model to improve the prediction of the drag force acting on the particle. The parallel computing library was developed for the data transfer between CFD and DEM solvers. The developed unresolved CFD and DEM solver was applied to a single sphere settling particle. The results obtained were compared with an empirical correlation and experimental data found in the scientific literature.

The present paper is organized as follows. Section 2 describes computational methods including governing equations for the unresolved CFD-DEM solver, the averaging method based on a kernel function, the volume fraction in the drag model, parallelization of the algorithm, and numerical methods used to discretize the set of governing equations. Section 3 presents the results and discussion for the single particle settlement benchmark. Finally, in Section 4, some conclusions are provided.

## 2. Computational Methods

### 2.1. Discrete Element Method (DEM)

DEM was proposed by Cundal and Strack [6] and simplifies the collision between particles to a linear spring-dashpot model. The motion of an individual particle was described by Newton's second law. The governing equation for translational motion of a particle was expressed as:

$$m_p \frac{du_p}{dt} = \sum f_{p,p} + \sum f_{p,w} + m_p g + f_{p,f} \tag{1}$$

where $m_p$ is the particle mass, $u_p$ is the particle translational velocity, and $f_{p,p}$ and $f_{p,w}$ are the contact force between particle and particle and the contact force between particle and wall, respectively. $m_p g$ is the gravitational force. $f_{p,f}$ represents the particle and fluid interaction force including the drag force ($f_d$), the buoyance force ($f_b$), the added mass force ($f_{AM}$), and the Basset force ($f_B$). To consider the added

mass force and the Basset force in an acceleration motion of the sphere, the integrated added mass coefficient ($C_A$) was employed [32]. The governing equation (Equation (1)) can then be expressed as:

$$\frac{\pi}{6}d_p^3\left(\rho_p + \rho_f C_A\right)\frac{du_p}{dt} = \sum f_{p,p} + \sum f_{p,w} + m_p g + f_{p,f} \tag{2}$$

where $d_p$ is the particle diameter, $\rho_p$ is the density of the particle, $\rho_f$ is the density of the fluid, $C_A$ is the integrated added mass coefficient which includes the added mass force and the Basset force of the falling particle, and $f_{p,f}$ represents the drag force ($f_d$) and the buoyance force ($f_b$).

The rotational motion equation of a particle was governed by the following equation:

$$I_p\frac{d\omega_p}{dt} = \sum T_t \tag{3}$$

where $I_p$ is the inertia moment of a particle, $\omega_p$ is the angular velocity of a particle, and $T_t$ is the torque due to the collision with the particle or the wall.

## 2.2. Computational Fluid Dynamics (CFD)

The fluid satisfies the mass and momentum conservation equations, and the motion of the particles was described by the volume fraction transport equation. The governing equations were expressed as:

$$\nabla \cdot u_f = 0 \tag{4}$$

$$\frac{\partial\left(\alpha_f u_f\right)}{\partial t} + \nabla \cdot \left(\alpha_f u_f u_f\right) = \frac{1}{\rho_f}\left(-\nabla p + \nabla \cdot \left(\alpha_f \tau_f\right) + \rho_f \alpha_f g + R_{f,p}\right) \tag{5}$$

$$\frac{\partial\alpha_f}{\partial t} + \nabla \cdot \left(\alpha_f u_f\right) = 0 \tag{6}$$

where $\alpha_f$ is the fluid volume fraction ($\alpha_f = 1 - \alpha_p$, where $\alpha_p$ is the particle volume fraction), $u_f$ is the fluid velocity, $p$ is the pressure, $\tau_f$ is the stress tensor of the fluid, $g$ is the gravity, and $R_{f,p}$ indicates the volumetric fluid and particle momentum exchange. $R_{f,p}$ could be expressed as:

$$R_{f,p} = K_{f,p}u_f - K_{f,p}\langle u_p\rangle \tag{7}$$

$$K_{f,p} = -\frac{\left|\sum f_{p,f}\right|}{V_c\left|u_f - \langle u_p\rangle\right|} \tag{8}$$

where $K_{f,p}$ is the fluid and particle interaction coefficient, $\langle u_p\rangle$ is the averaged particle velocity based on each cell, and $f_{p,f}$ is the forces acting on the particles by the fluid. $\alpha_f$, $\langle u_p\rangle$, and $f_{p,f}$ were expressed as:

$$\alpha_f = 1 - \frac{1}{V_c}\sum_{k=1}^{N_p} w_{p,k}V_{p,k} \tag{9}$$

$$\langle u_p\rangle = \frac{1}{\alpha_p V_c}\sum_{k=1}^{N_p} w_{p,k}u_{p,k}V_{p,k} \tag{10}$$

$$f_{p,f} = \frac{1}{V_c}\sum_{k=1}^{N_p} w_{p,k}(f_{d,k} + f_{b,k}) \tag{11}$$

where $N_p$ is the number of the particles that contribute to the fluid volume fraction of a cell, $V_{p,k}$ is the volume of the $k^{\text{th}}$ particle, and $V_c$ is the volume of a single cell. $w_{p,k}$ is a weighting factor of the $k^{\text{th}}$

particle to a cell. The weighting factor could be expressed using various functions, but the kernel function was selected in this study. In $f_{p,f}$, the drag force ($f_d$) and buoyance force ($f_b$) (or the Archimedes buoyance force) were considered as the main interaction forces [30]. The drag force ($f_d$) was calculated as:

$$f_d = \frac{1}{8} C_d \rho \pi d_p^2 a_f^{-(\chi+1)} (u_f - u_p) |u_f - u_p| \tag{12}$$

where the drag coefficient ($C_d$) [33] and the drag force model expression ($\chi$) [34] were expressed as:

$$C_d = \frac{24}{(9.06)^2} \left( \frac{9.06}{\sqrt{Re_p}} + 1 \right)^2 \tag{13}$$

$$\chi = 3.7 - 0.65 exp \left[ -\frac{(1.5 - \log Re_p)^2}{2} \right] \tag{14}$$

where $d_p$ is the particle diameter and $C_d$ is the drag coefficient, which is a function of the Reynolds number based on the particle diameter. $a_f^{-(\chi+1)}$ is the pre-factor that describes the presence of neighboring particles in a dense particle-laden flow. It meant that the drag model depended on the volume fraction. When the cell size was larger than the particles, the dependence of the volume fraction in the drag model could be negligible for a very dilute flow [5,22]. The drag force acting on a particle uses the relative velocity ($|u_f - u_p|$) at the particle center. The particle's Reynolds number ($Re_p$) was expressed as:

$$Re_p = \frac{\rho d_p |u_f - u_p|}{\mu} \tag{15}$$

In this study, the interpolated fluid velocity at each particle center [35] was applied to the drag model. The interpolated fluid velocity ($\widetilde{u}_f$) at the particle center can be expressed as:

$$\widetilde{u}_f = \sum_{m=1}^{N_c} w_{p,m} u_{f,m} \tag{16}$$

where $N_c$ is the total number of cells that contains the particles.

### 2.3. Kernel-Based Averaging Method

The velocity and pressure from CFD were mapped to the particles using an averaging method with a weighting function. The weighting factor of the particles was calculated based on the volume fraction of the particle in each cell. The centered volume fraction method and the divided volume fraction method are approaches that have been widely used [31]. In the centered volume fraction method, the weight is decided based on whether the particle center was located in a cell or not. The weighting factor of the particles in the cell was zero or one. On the other hand, in the divided volume fraction method, the weighting factor is calculated using the divided particle volume fraction by the cell. In several cases, the divided volume fraction method provided more reasonable results [30]. However, both methods had limitations when the cell size was close to or smaller than the particle diameter [21,31]. The fine cells around a boundary layer and a complex fluid flow caused a computational instability [16,18]. An averaging method using the kernel function has been suggested [19,24,35,36]. When the particle diameter was close to the cell size, the Gaussian kernel function played an important role in the computational stability and accuracy of the solutions [37]. The computational cost increased in proportion to the number of particles and the total number of cells [35]. To reduce the computational cost, a support region of the kernel functions was considered [35,36]. Figure 1 shows four cells and one single particle ("A") in the two-dimensional Cartesian coordinate. The support region of particle "A" was defined as the circle of radius ($R_s$). The center of the support region coincided with

the particle center, and the radius could be expressed in proportion to the particle diameter ($R_s = Kd_p$). $K$ was a constant and was set to 3 considering the computational cost and accuracy [35]. The cells were influenced by particle "A" when the cell center was included in the radius. The influence was represented by the weighting factor ($w_A$) with the kernel function. The weighting factor of particle "A" and the Gaussian kernel function ($g_{(x-x_m)}$) could be expressed as:

$$w_{A,c} = \frac{g_{(x-x_c)} V_c}{\sum_{m=1}^{N_c} g_{(x-x_m)} V_m} \tag{17}$$

$$g_{(x-x_m)} = \begin{cases} \frac{1}{(b\sqrt{2\pi})^3} exp(-\frac{|x-x_m|^2}{2b^2}), & |x - x_m| < R_s \\ 0, & |x - x_m| \geq R_s \end{cases} \tag{18}$$

where $N_c$ is the total number of the cells in the radius of the particle "A" ($N_c$ = 3 in Figure 1). The Gaussian kernel function ($g_{(x-x_m)}$) is expressed as a function of the distance between the cell center and the particle center ($|x - x_m|$) in the support region. $b$ indicates the bandwidth of the kernel function and can be expressed as $b = kd_p$. The particle weighting factors in the kernel function satisfied the following normalization condition [24]:

$$\sum_{m=1}^{N_c} w_{A,m} = 1 \tag{19}$$

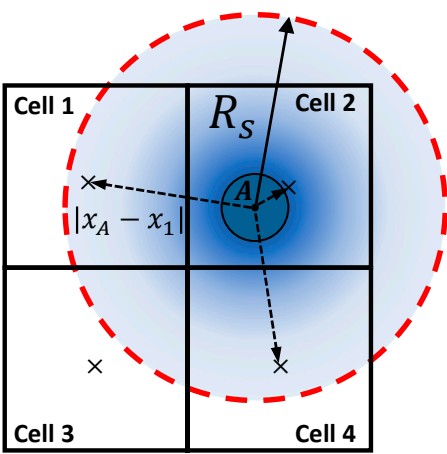

**Figure 1.** Cells, particle, and support region.

## 2.4. Volume Fraction in Drag Model

The volume fraction of the cell, where the particle center was located, was smaller than one. Thus, the drag force of the particle was predicted to be smaller than the actual drag force [3]. Revised drag force prediction methods have been proposed to improve the accuracy of the drag force prediction, and also to obtain computational stability for several cell sizes [5]. The revised volume fraction was applied to the drag force model. The revised volume fraction was expressed as:

$$\widetilde{\alpha_f} = 1 - \frac{V_p}{E_p \sum_{m=1}^{N_c} V_{c,m}} \tag{20}$$

where $N_c$, $V_p$, and $E_p$ denote the total number of cells included in the support region of the particle, the volume of the particle, and the expansion coefficient of the particle volume, respectively.

When the expansion coefficient was one, the volume fraction was the same as when the "big particle" volume fraction method [38] was used. The revised volume fraction was applied to the drag force model.

### 2.5. Parallelization of Computing Resources

To consider the parallelization of the computing resources, it was necessary to define neighboring cells located in the support region in order to calculate the weighting factor of the particles. When the support region of the particle was included in two or more computing processors, the data transfer between the processors should be considered. Figure 2 shows the support region of the particle and the interface between two processors. The red dotted line shows the interface between two processors. In the support region, the cell center in the processor [0] was represented by the hollow circle, while the cell center in the processor [1] was represented by the hollow square. The cell center beyond the support region was shadowed. The center of particle "A" was located in the cell number 5 in the processor [0]. The mapping of particle "A" to the CFD cells was performed through the following steps:

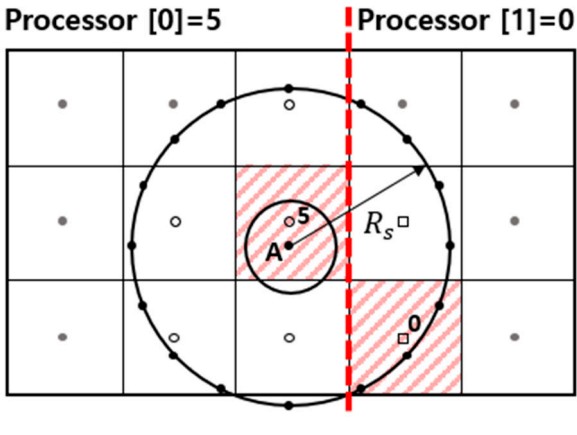

(**a**) Detecting cells.

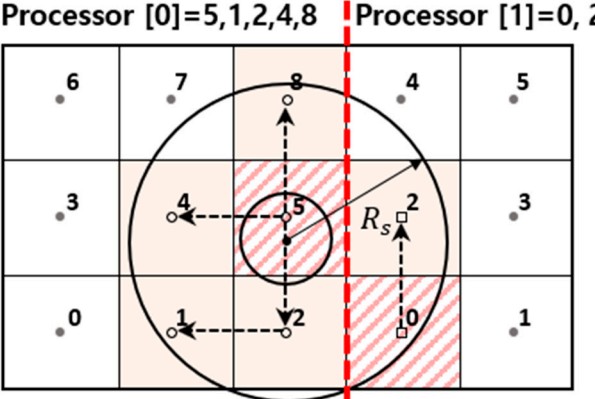

(**b**) Listing sub-cells.

**Figure 2.** *Cont.*

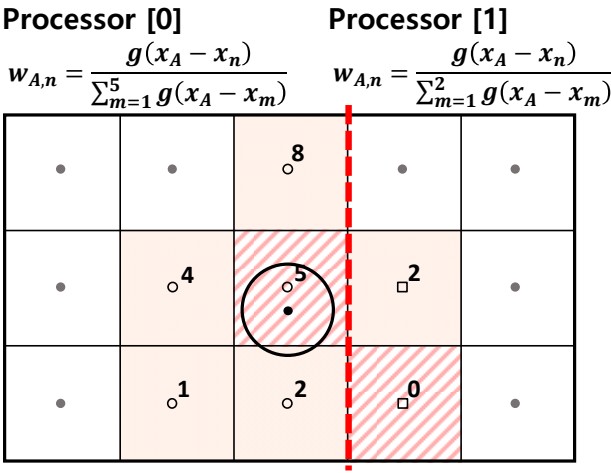

(**c**) Calculating weighting factors.

**Figure 2.** Data transfer between two processors. (**a**) Detecting the cells; (**b**) Listing sub-cells; (**c**) Calculation of the weighting factors.

(i) In Figure 2a, the cells in the particle "A" of each processor were marked. Cell number 5 in processor [0] and cell number 0 in processor [1] were selected.

(ii) A sub-cell list in the support region was created in each processor in Figure 2b. The sub-cell list in processor [0] was 5, 1, 2, 4, and 8, while the sub-cell list in processor [1] was 0 and 2.

(iii) For the data transfer between sub-cells, the weighting factors were calculated in Figure 2c.

(iv) Calculations of $\alpha_p$, $u_p$ and $f_d$ using the weighting factors were carried out.

The cells for which the center was not located in the support region were deactivated in the prediction of the drag and buoyancy forces.

The single particle settling was simulated to validate the parallelization. The particle diameter and density were 15 mm and 1120 kg/m$^3$, respectively. The density of fluid was 965 kg/m$^3$ and the viscosity was 0.00212 Ns/m$^2$. Figure 3 shows the settling velocity. The particle passed the interface between two processors at around one second. When data transfer was taken into account, non-physical fluctuation of the setting velocity disappeared at the process interface.

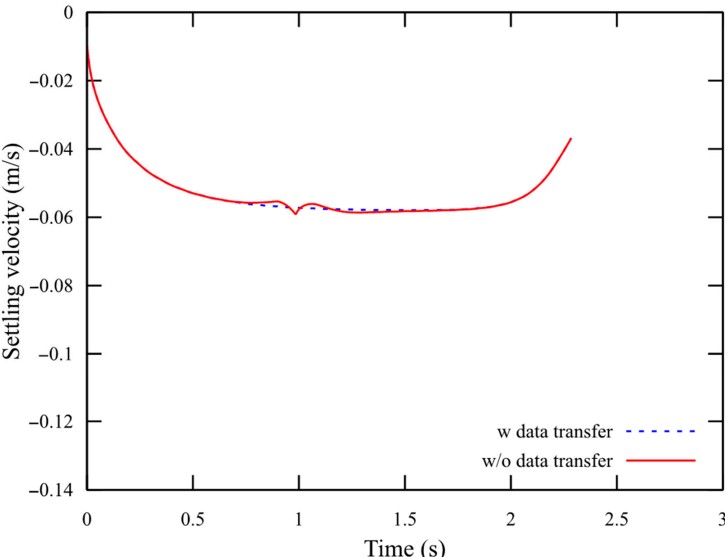

**Figure 3.** Data transfer between processors.

### 2.6. Numerical Methods

The time derivative terms were discretized using a second-order accurate implicit scheme. The gradient at the cell centers was calculated using the Euler method. The convection terms were discretized using a second-order accurate total variation diminishing (TVD) scheme with the Sweby limiter [39]. The velocity and pressure coupling, and overall solution procedure, were based on the pressure implicit with the splitting of the operator (PISO) algorithm [40]. All simulation cases were performed under laminar flow conditions. The discretized algebraic equations for the pressure and velocity were solved using a pointwise Gauss-Seidel iterative algorithm, while an algebraic multi-grid method [41] was employed to accelerate solution convergence [42].

The time step in the CFD solver was set to be bigger than that in the DEM solver [21]. The time steps for the CFD and DEM solvers were $1 \times 10^{-3}$ s and $1 \times 10^{-5}$ s, respectively. Every one-hundred steps, both the CFD and DEM results were coupled. The stability was dominated by the DEM solver. The time step in the DEM solver was set to be 20% smaller than the Rayleigh critical time step [43]. The simulation time was advanced when the normalized residuals for the solutions had dropped by six orders of magnitude.

## 3. Results and Discussion

To verify and validate the developed unresolved CFD and DEM coupled solver, several cases of falling solid spheres were considered.

### 3.1. Free Settling of a Sand Particle in Water

To verify the developed solver and numerical methods, the settling velocity of free-falling sand particles in a water tank was considered. The computed settling velocity was compared with that of an empirical correlation [33]. The computational domain size was 25 $d_p$ long, 25 $d_p$ wide, and 75 $d_p$ high, as shown in Figure 4. Here, $d_p$ is the particle diameter, and the cell size ($\Delta x$) is 1 $d_p$. Table 1 shows the test cases. Sand particle diameters of 0.125, 0.25, 0.5, 1.0, and 2.0 mm were considered. The density and viscosity of the water were 998.25 kg/m$^3$ and $1.002 \times 10^{-3}$ Ns/m$^2$, respectively. The Reynolds numbers of each particle were 1 to 516 based on the settling velocity [33]. The particles fell freely from a standstill at a height of 74 $d_p$. The influence of the bandwidth ($b$) and cell size ratio ($\Delta x / d_p$) was investigated. The three bandwidths (1, 3, and 6 $d_p$) and two cell size ratios ($\Delta x / d_p = 1$ and 4) were considered. Figure 5 shows the velocity and pressure contours of the falling particle. The effect of the particle on the fluid flow in the two-way coupled solver was clearly identified as the cell size approached the particle size. Local flow disturbances occurred within the support region of the particle. Figure 6 shows the settling velocities based on the averaged kernel method, divided method, and empirical correlation [33]. In the cell size ratio ($\Delta x / d_p$) of 4, the averaging kernel method showed good agreement with the result according to the divided method and empirical correlation. The influence of the bandwidth could not be observed because the support region ($R_s$) was smaller than the cell size ($\Delta x$). Figure 6b shows the results with the cell size ratio ($\Delta x / d_p$) of 1. Because the support region ($R_s$) was bigger than the cell size ($\Delta x$), the bigger bandwidth showed a converged result. It was explained by the nature of the two-way coupling between the Eulerian and Lagrangian frameworks. The relative velocity in the drag was a dominant variable. When a particle fell from stationary fluid, the momentum transferred from the particle to the fluid flow created a non-zero fluid velocity component in the negative vertical direction. When estimating the drag force using the fluid velocity interpolated to the particle, it underestimated the relative velocity of the particle and thus the drag force. As bandwidth increased in the given support region, the effect of the momentum exchange logically mitigated. The bigger bandwidth, the better the drag was predicted. The result with the divided method showed a numerical wiggle, and a difference from the empirical correlation. As the particle size was close to the cell size, the fluid volume fraction was not accurate. This inaccurate fluid volume fraction affected the fluid velocity in the drag force model. Figure 7 shows

the drag model variables ($C_d$, $a_f$, $|u_f - u_p|$, $u_p$, and $u_f$) based on the divided method. In the results with a cell size ratio of 4, the variables showed stable behavior. At a cell size ratio of 1, the fluid volume fraction of the cell fluctuated greatly, and the average value decreased. The particle volume fraction increased as the particle size became closer to the cell size. This caused a difference in the fluid volume fraction between the owner and neighbor cells. The fluctuation of the fluid volume fraction caused discontinuous pressure and velocity fields [16]. The underestimated fluid volume fraction indicated a decrease of fluid mass and an increase of fluid velocity. The drag force was underestimated with a decrease in the relative velocity. The underestimated fluid volume fraction indicated a decrease of the fluid mass and an increase of the fluid velocity. The settling velocity decreased with a decrease in the relative velocity and an increase of the drag coefficient. As a result, the settling velocity fluctuated when the particle passed the cell interface [18,22]. Figure 8 shows the drag model variables according to the averaging kernel method with the cell size ratio ($\Delta x/d_p$) of 1. In the averaging kernel method, the variables of the drag model were calculated stably regardless of the cell size ratio. The results showed that computational stability could be obtained, and the influence of the cell size could be minimized when the support region was bigger than the cell size and $b > 3\,d_p$. Based on these results, the cell size ratio ($\Delta x/d_p$) = 1 and $b = 6\,d_p$ were selected.

$$L \times W \times H = 25\,d_p \times 25\,d_p \times 75\,d_p$$

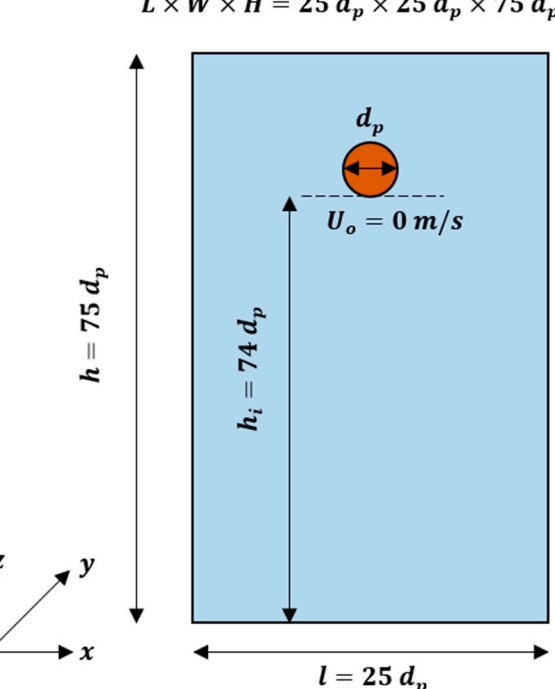

**Figure 4.** Description of single sphere settling.

**Table 1.** Test cases.

| Case Number | Diameter of Particle (mm) | Density of Sand (kg/m³) | Density of Water (kg/m³) | Viscosity of Water (Ns/m²) | Reynolds Number (-) |
|---|---|---|---|---|---|
| 1 | 2 | 2463 | 998.25 | 0.001002 | 516.1 |
| 2 | 1 | 2488 | 998.25 | 0.001002 | 146.5 |
| 3 | 0.5 | 2523 | 998.25 | 0.001002 | 33.7 |
| 4 | 0.25 | 2571 | 998.25 | 0.001002 | 7.8 |
| 5 | 0.125 | 2494 | 998.25 | 0.001002 | 1.3 |

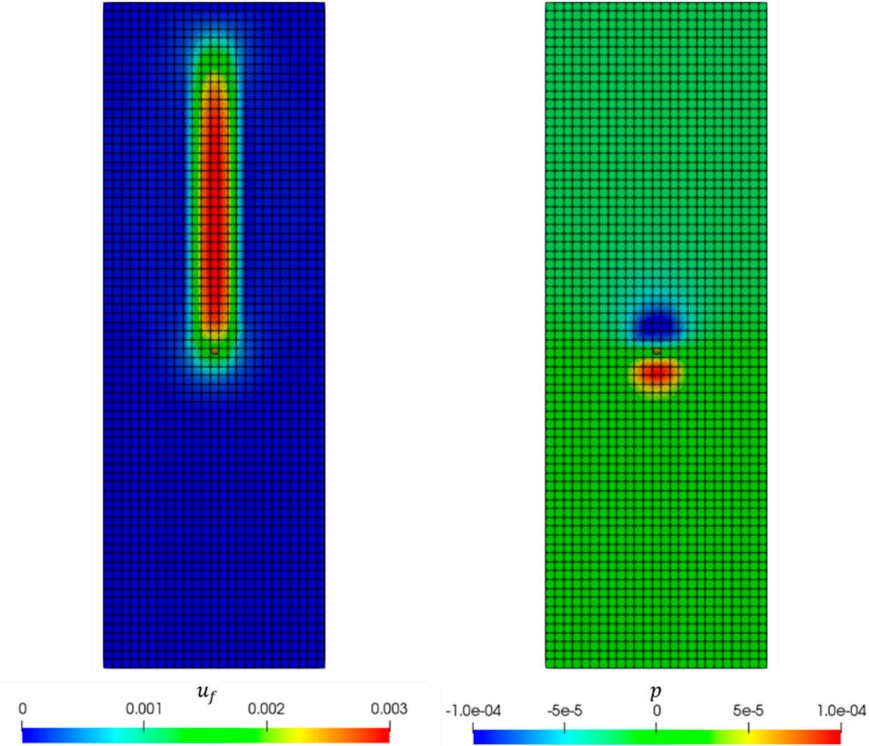

**Figure 5.** Velocity ($u_f$) and pressure ($p$) contours ($\Delta x/d_p = 1$).

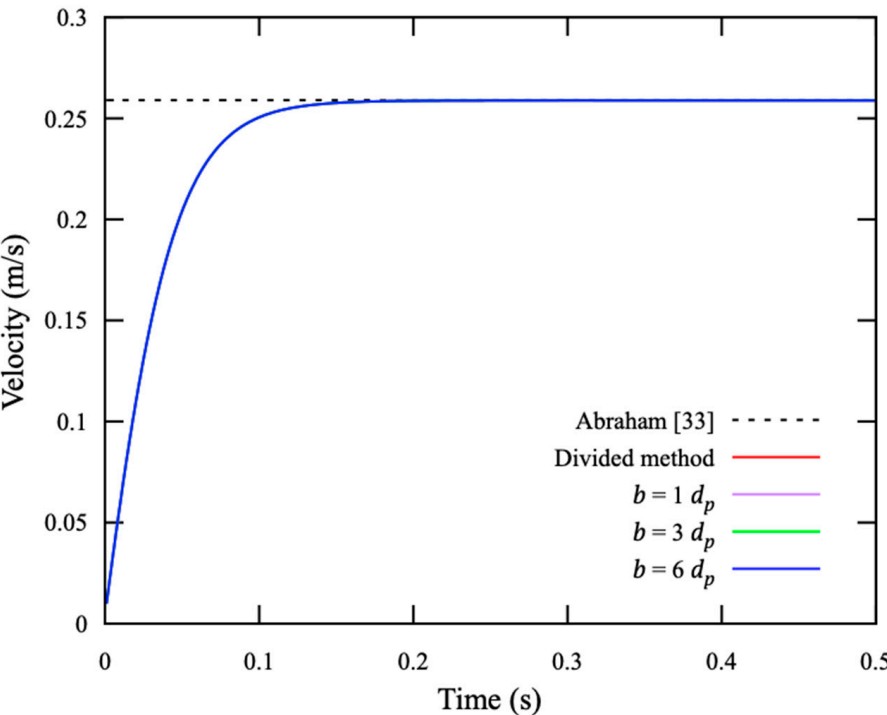

(**a**) Cell size ratio ($\Delta x/d_p$) = 4.

**Figure 6.** *Cont.*

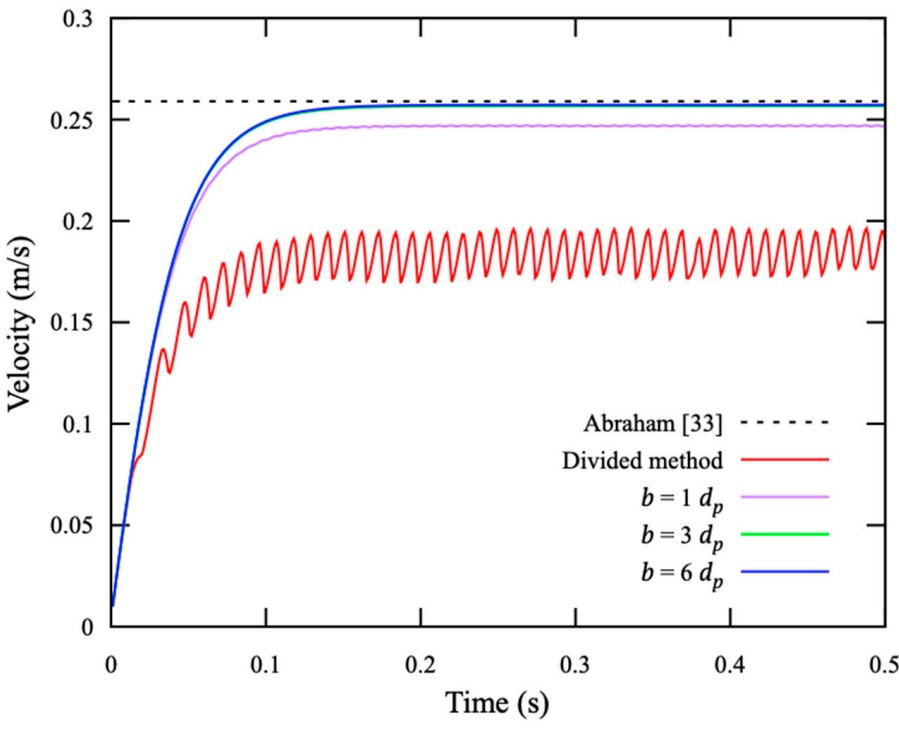

(**b**) Cell size ratio ($\Delta x/d_p$) = 1.

**Figure 6.** Settling velocity for various bandwidths (*b*) and cell size ratios ($\Delta x/d_p$). (**a**) Cell size ratio ($\Delta x/d_p$) = 4; (**b**) Cell size ratio ($\Delta x/d_p$) = 1.

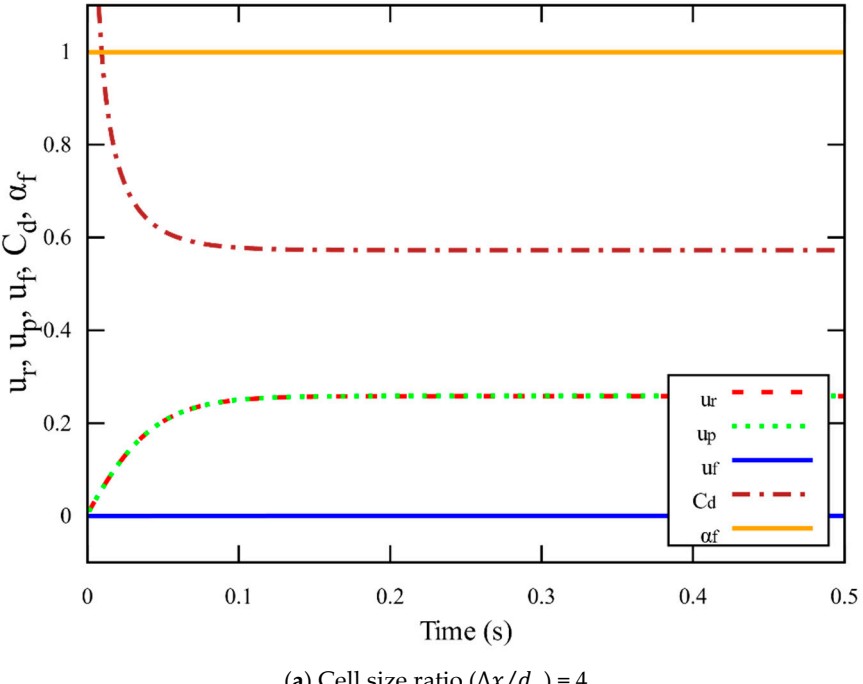

(**a**) Cell size ratio ($\Delta x/d_p$) = 4.

**Figure 7.** *Cont.*

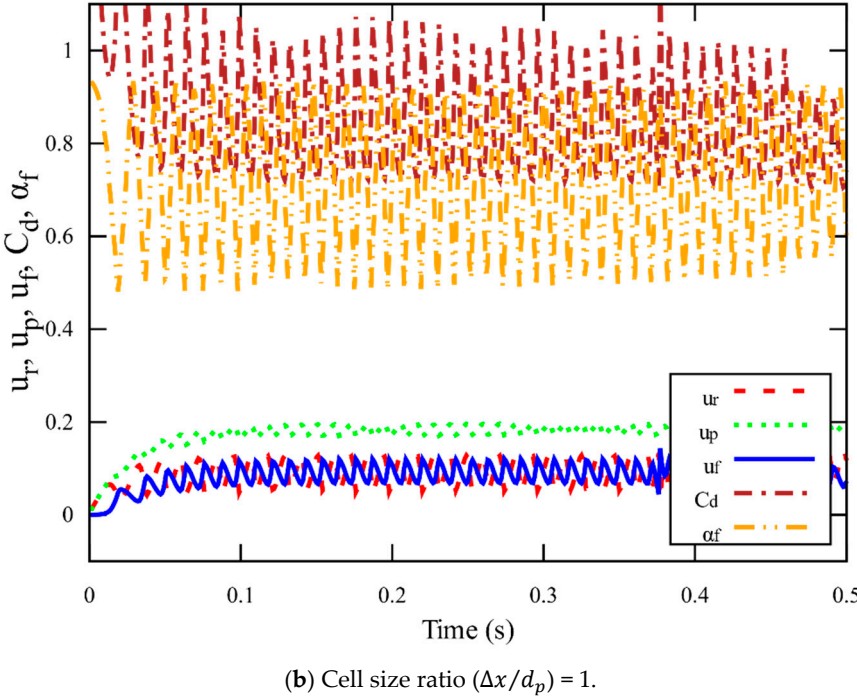

(**b**) Cell size ratio ($\Delta x / d_p$) = 1.

**Figure 7.** Drag model variables in the divided method. (**a**) Cell size ratio ($\Delta x / d_p$) = 4; (**b**) Cell size ratio ($\Delta x / d_p$) = 1.

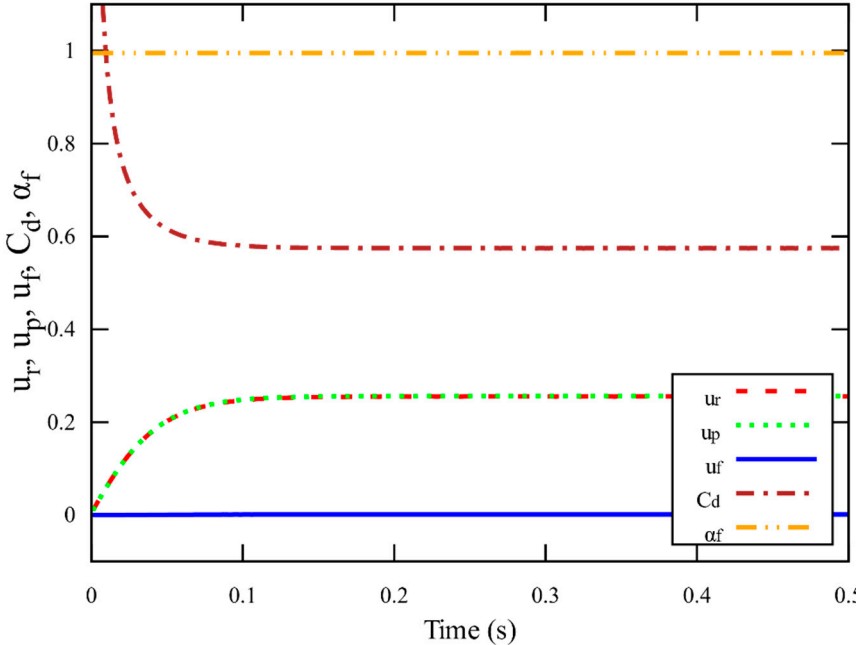

**Figure 8.** Drag model variables in the averaging kernel method with $\Delta x / d_p = 1$

Table 2 lists the settling velocity for different particle diameters. The velocities showed good agreement with the empirical correlation [33]. The particle diameter decreased with an increase in the difference. The drag force was underestimated because the relative velocity was underestimated when the Reynolds number for the particle was close to the Stokes flow. It meant that the significant momentum exchange from the particle to the fluid was occurred as the particle Reynolds number was closed to the Stokes flow [5].

**Table 2.** Settling velocity.

| Case Number | Particle Diameter (mm) | Reynolds Number (-) | Settling Velocity (m/s) | | Difference (%) |
|---|---|---|---|---|---|
| | | | Empirical Correlation [33] | Present | |
| 1 | 2 | 516.1 | 0.2590 | 0.2563 | 1.04 |
| 2 | 1 | 146.5 | 0.1480 | 0.1467 | 0.88 |
| 3 | 0.5 | 33.7 | 0.0742 | 0.0740 | 0.27 |
| 4 | 0.25 | 7.8 | 0.0313 | 0.0318 | −1.60 |
| 5 | 0.125 | 1.3 | 0.0101 | 0.0109 | −7.92 |

### 3.2. Free Settling of a Steel Sphere in Water

The settling velocity of a free-falling steel sphere in a water tank was simulated at a high particle Reynolds number. The computational domain size was 115 mm long, 30 mm wide, and 280 mm high. The particle fell freely from the top of the tank with no initial velocity. The particle diameter and density were 3.18 mm and 7820 kg/m$^3$, respectively [44]. The density of water was 1000 kg/m$^3$ and the viscosity was 0.001 Ns/m. The Reynolds number ($Re = \frac{\rho d_p |u_f - u_p|}{\mu}$) based on the particle diameter was 2440. The cell number was 33 × 9 × 88. The cell size ratio is close to one. Figure 9 shows the settling velocity of the particle and the falling height. The results were compared to those obtained using the divided method and the experimental data [44]. The results obtained using the averaging kernel method were very close to the experimental results. On the other hand, the results obtained using the divided method oscillated and differed greatly from the experimental data. In the results obtained using the divided method, the increased drag in the cell resulted in a slow settling velocity, and the difference of the volume fraction between the adjacent cells caused the oscillation.

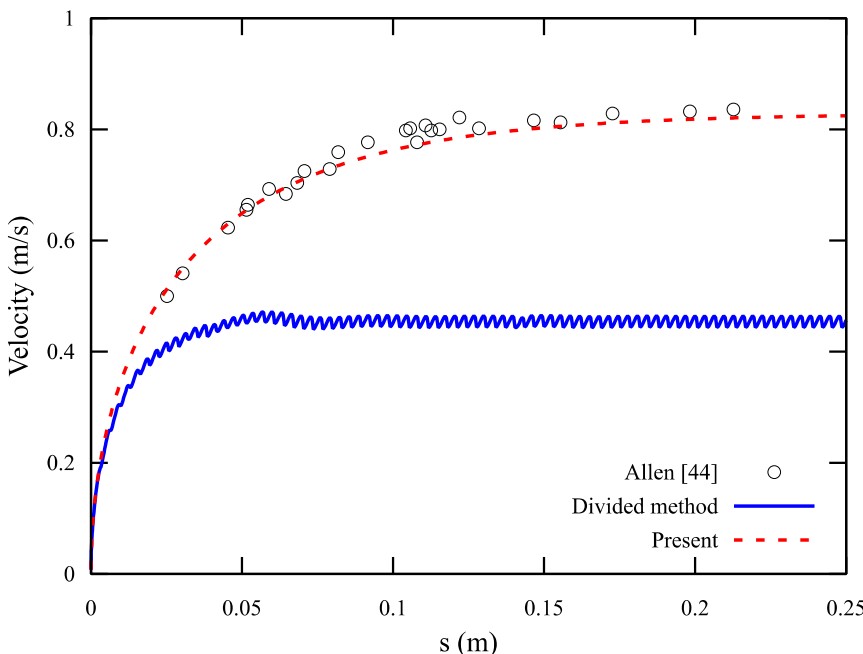

**Figure 9.** Particle settling velocity and height relationship.

### 3.3. Free Settling of a Glass Sphere in Water

The influence of the integrated added mass coefficient and particle volume expansion coefficient ($E_p$) on the settling velocity was investigated. The computational domain extent was 1.25 m long, 1.25 m wide, and 10 m high. The diameter and density of the particle were 0.167 m and 2567.7 kg/m$^3$,

respectively. The density of water was 1000 kg/m$^3$ and the viscosity was 0.005416 Ns/m. The particle fell freely from a height of 9 m, and the Reynolds number was 360 [45]. The cell number was $15 \times 15 \times 56$. The cell size ratio was close to one.

Figure 10 shows the influence of the integrated added mass coefficients ($C_A$) with the particle volume expansion coefficient ($E_p$) of 1.0. The computational results were dimensionless for the time and velocity, and were expressed as:

$$t_{ref} = \sqrt{\frac{d_p}{g}} \tag{21}$$

$$u_{ref} = \sqrt{d_p g} \tag{22}$$

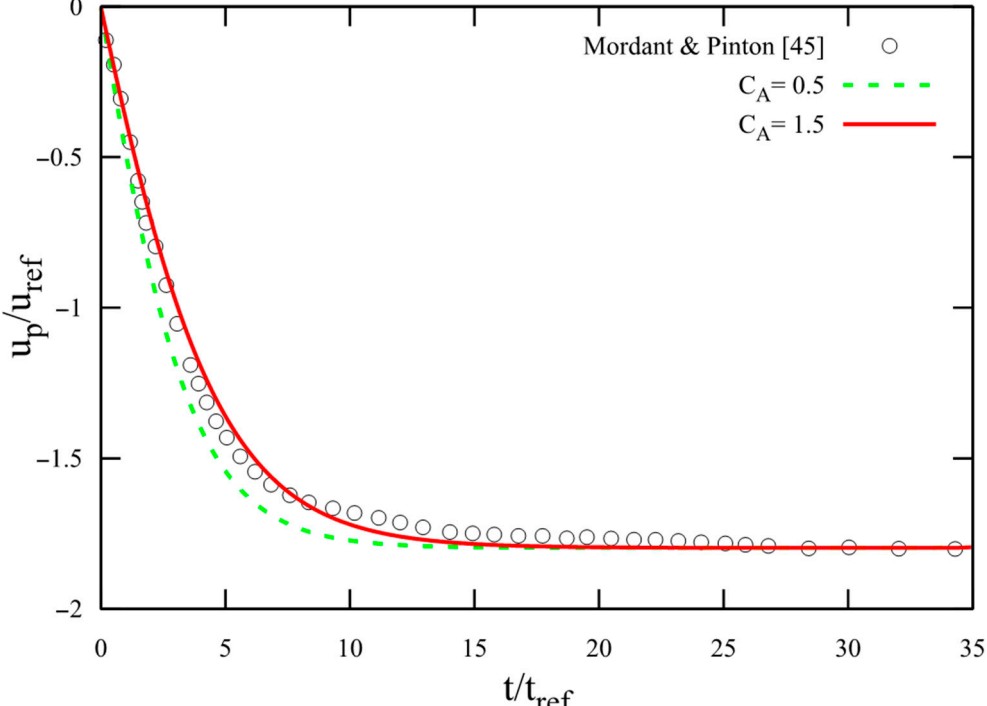

**Figure 10.** Influence of several integrated added mass coefficients when using a particle volume expansion coefficient of 1.0.

The integrated added mass coefficient of 1.5 showed good agreement with the experimental data [29]. For the integrated added mass coefficient of 0.5, the particle velocity was overpredicted because the Basset force was ignored or there was a drag prediction error in the transition period [32]. The integrated added mass coefficient influenced the transition period. The particle settling velocity in the steady-state period had no relation with the integrated added mass coefficient.

Figure 11 shows the influence of the particle volume expansion coefficients ($E_p$) with the integrated added mass coefficient ($C_A$) of 1.5. As the particle volume expansion coefficient decreased, the drag force increased, indicating that the particle motion quickly reached steady state. The decrease of the particle volume expansion coefficient indicated a relatively bigger particle volume fraction in Equation (20).

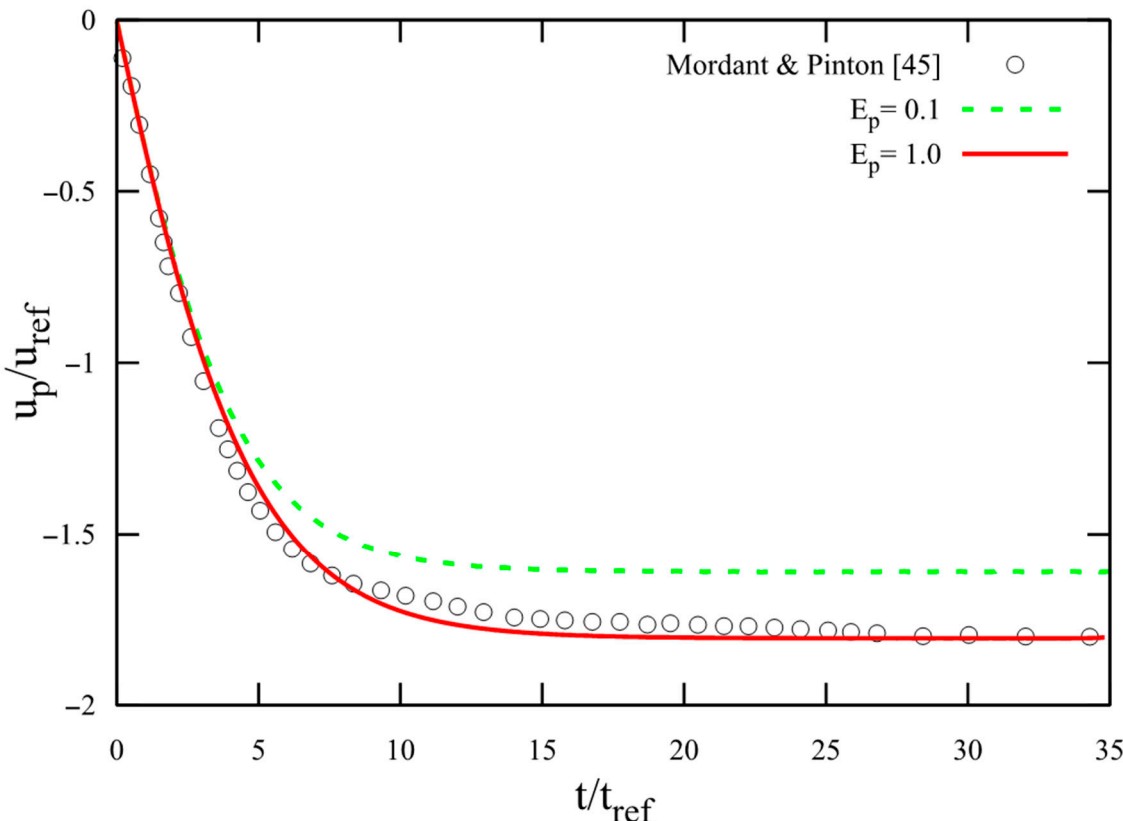

**Figure 11.** Influence for of several particle volume expansion coefficients when using an integrated added mass coefficient of 1.5.

### 3.4. Free Settling of a Nylon Sphere in Silicon Oil

A nylon sphere falling in a closed square box filled with silicone oil was studied [26]. The square box size was 100 mm long, 100 mm wide, and 160 mm high. The diameter and density of the nylon sphere were 15 mm and 1120 kg/m$^3$, respectively. The sphere fell from a height of 120 mm above the bottom surface. Table 3 lists the simulation cases. The Reynolds number was based on the steady-state settling velocity of the sphere ($u_\infty$). Here, $u_\infty$ was determined using the empirical correlation [33]. The domain cell divisions were $7 \times 7 \times 10$ for length, width and height, respectively. The cell size ratio was close to one.

**Table 3.** Test cases.

| Case Number | Density of Fluid (kg/m$^3$) | Viscosity of Fluid (Ns/m$^2$) | Reynolds Number (-) | $u_{max}/u_\infty$ |
|:---:|:---:|:---:|:---:|:---:|
| 1 | 970 | 0.00373 | 1.5 | 0.947 |
| 2 | 965 | 0.00212 | 4.1 | 0.953 |
| 3 | 962 | 0.00113 | 11.6 | 0.959 |
| 4 | 960 | 0.00058 | 31.9 | 0.955 |

The influence of the integrated added mass coefficient on the unsteady-state motion of the falling particle was studied for the Reynolds number of 31.9. Figure 12 shows the variation of the height ($h/d_p$) normalized by the particle diameter and the settling velocity of the particle. In the case where the integrated added mass coefficient was ignored, the particle reached the bottom surface earlier than in the experimental data [26] because the acceleration of the particle in the transient region was overpredicted. Therefore, by considering the integrated added mass coefficient, it was possible to

accurately predict the unsteady drag of the particle and the integrated added mass force, and as such, the trajectory of the particle was in good agreement with the experimental data.

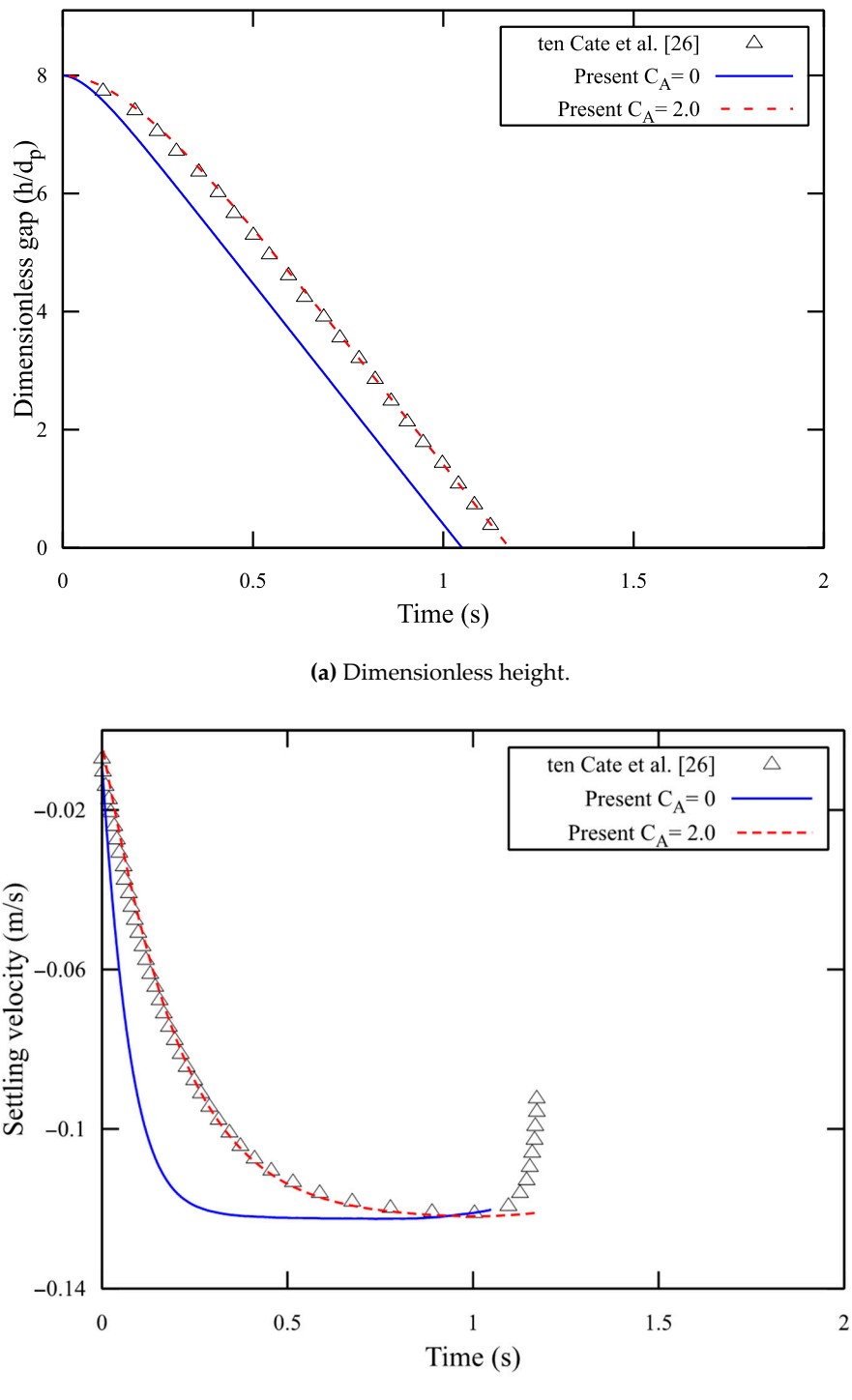

**(a)** Dimensionless height.

**(b)** Settling velocity.

**Figure 12.** Particle trajectory and settling velocity. (**a**) Dimensionless height; (**b**) Settling velocity.

Figure 13 shows the settling velocity for the four Reynolds numbers (*Re* = 1.5, 4.1, 11.6, and 31.9) with the integrated added mass coefficient of 2.0. The solid lines represent the present simulation results, and the symbols indicate the experimental results [36]. As can be seen from the experimental results, the velocity of the sphere gradually increased until it reached steady state. As the sphere approached

the bottom of the box, it began to slow down due to the repulsive force caused by the squeezing motion of the fluid in the gap between the sphere and the box's bottom wall [26,27,46]. To simulate the repulsion force, a very fine cell or lubrication theory should be considered [38]. However, the effect of the bottom wall was ignored because it was not the principal subject of this paper. That is why the simulation results were shown until the particle touched the bottom wall. The settling velocity in the transition and steady-state regions showed good agreement with the experimental data [26] and other simulations [27,38,46–50]. Table 4 shows the velocity ratios ($u_{max}/u_\infty$). The maximum settling velocity ($u_{max}$) was normalized by the steady-state terminal velocity ($u_\infty$) in an infinite medium. The empirical correlation [33] was used to determine the steady-state terminal velocity ($u_\infty$) [26]. In the experiment, the velocity ratio ($u_{max}/u_\infty$) was lower than one because the particle moved at a velocity lower than $u_\infty$ due to the wall effect [36]. The simulation results for the normalized settling velocity were within 1% of the experimental results.

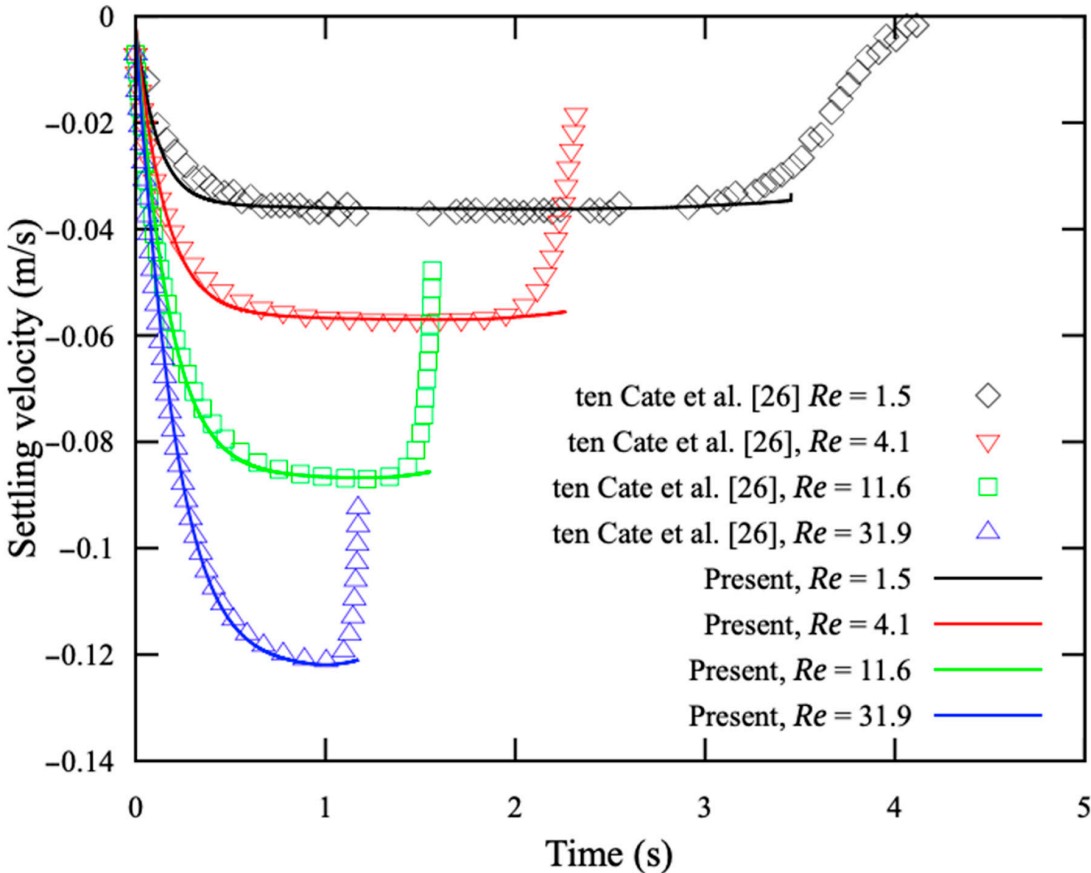

**Figure 13.** Settling velocity for different Reynolds numbers.

**Table 4.** Velocity ratio ($u_{max}/u_\infty$) for different Reynolds numbers.

| | Velocity Ratio ($u_{max}/u_\infty$) | | | |
|---|---|---|---|---|
| **Case** | **1** | **2** | **3** | **4** |
| Reynolds number | 1.5 | 4.1 | 11.6 | 31.9 |
| Present | 0.943 | 0.949 | 0.955 | 0.949 |
| Experiment [36] | 0.947 | 0.953 | 0.959 | 0.955 |
| Difference (%) | 0.41 | 0.33 | 0.39 | 0.65 |

## 4. Conclusions

In this study, simulations for a single particle settling were carried out using the newly developed unresolved CFD and DEM coupled solver. The solver used was developed by coupling open-source computational fluid dynamics libraries, termed OpenFOAM, with discrete element method libraries, termed LIGGGHTS [31]. The kernel-based averaging method was applied within the developed solver, and the revised volume fraction was introduced into the drag model. A parallel computing library was developed for the CFD and DEM coupling.

To verify and validate the developed solver, the settling velocities of a single particle falling were simulated and compared with empirical correlation and experimental data. In the divided method, the predicted drag force was inaccurate for a cell size ratio ($\Delta x / d_p$) of about one. On the other hand, the results obtained using the kernel-based averaging method predicted the settling velocities within 2% of the empirical correlation at the range from the low to medium Reynolds numbers. Where the integrated added mass coefficient was greater than 0.5, it could be seen that in addition to the integrated added mass force, the unsteady drag force was increased. As the particle volume expansion coefficient decreased, the hydrodynamic drag increased due to the increased particle volume fraction in the drag model. In the simulation of the nylon sphere with selected numerical methods, the results of the sphere trajectory and the settling velocity for different Reynolds numbers were in good agreement with the experimental data.

**Author Contributions:** Unresolved CFD and DEM coupled solver development and computations, S.S.; writing-original draft preparation, S.S. and S.P.; writing-review and editing S.S. and S.P.; supervision S.P. All authors have read and agreed to the published version of the manuscript.

**Funding:** This research was supported by the National Research Foundation (NRF-2018R1A1A1A05020799), a division of the government of Korea.

**Conflicts of Interest:** The authors declare no conflict of interest.

## Nomenclatures

$b$:        Bandwidth for the kernel function
$C_A$:    Integrated added mass coefficient
$C_d$:    Drag coefficient
$d_p$:    Particle diameter
$E_p$:    Particle volume expansion coefficient
$f_b$:    Buoyancy force
$f_d$:    Drag force
$f_{p,f}$:    Fluid force acting on the particle
$f_{p,p}$:    Force between particle and particle
$f_{p,w}$:    Force between particle and wall
$g$:        Gravity
$I$:        Moment of inertia of the particle mass
$K_{f,p}$:    Fluid and particle interaction coefficient
$m_p$:    Particle mass
$N_p$:    Total number of particles
$p$:        Static pressure
$Re_p$:    Particle Reynolds number
$R_{f,p}$:    Momentum exchange between fluid and particle
$R_s$:    Support region for particle
$t$:        Time
$T_p$:    Torque according to contact between particle and particle, or particle and wall
$u_f$:    Fluid velocity
$\widetilde{u}_f$:    Averaged fluid velocity at the particle center
$u_p$:    Particle velocity

| $\langle u_p \rangle$: | Averaged particle velocity based on a cell |
| $V_{p,k}$: | Volume of $k^{\text{th}}$ particle |
| $V_c$: | Volume of a single cell |
| $w$: | Angular velocity of the particle |
| $w_{p,k}$: | Weighting factor of $k^{\text{th}}$ particle |
| $\alpha_f$: | Fluid volume fraction |
| $\alpha_p$: | Particle volume fraction |
| $\mu$: | Viscosity coefficient |
| $\rho_f$: | Fluid density |
| $\rho_p$: | Particle density |
| $\tau$: | Viscous shear stress tensor |

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
