# Peer review of "Unresolved CFD and DEM Coupled Solver for Particle-Laden Flow and Its Application to Single Particle Settlement"

_jmse, doi:10.3390/jmse8120983_

Round 1

Reviewer 1 Report

The final revision improves the manuscript quality, thus I recommend to accept the manuscript.

Author Response

The authors appreciate the reviewer’s favorable review and comments.

Reviewer 2 Report

The manuscript should be revised according to the comments in the attached file.

Author Response

The authors appreciate the reviewer’s favorable review and constructive suggestions. The authors fully accepted all your suggestions in the attached file and revised the paper. The authors highlighted all changes in the revision. The authors feel that our paper has improved.

This manuscript is a resubmission of an earlier submission. The following is a list of the peer review reports and author responses from that submission.

Round 1

Reviewer 1 Report

In this manuscript, the authors study the settling of individual particles using an Euler-Lagrange formalism. The main feature of their framework resides in the operator that is used to spread information from the Lagrangian particles back onto the Eulerian grid: it consists in a convolution with a (clipped) Gaussian kernel. They consider various experimental settling experiments, and compare their numerical results to the findings of these experimental studies.

The use of convolution with a Gaussian kernel, as a mean to compute the volume-fraction and momentum exchange term associated with a Lagrangian particle, is a relatively recent and active topic of research. Unfortunately, I struggle to see the relevance of the work presented in this manuscript, and I must question its scientific soundness.

I have identified many conceptual flaws in the paper - I list of few of them here:
- First of all, the framework presented in the manuscript seems like a total overkill for the purpose of the study, which is to look at the very simple case of a settling particle. Moreover, it is used in a very unconventional, and presumably wrong, way.
- The governing Navier-Stokes equations (4-6), are given in their geometrically filtered version, including the contribution of volume fraction, which is relevant for the study dense particle-laden flows, and completely inappropriate for dilute (1 particle) flow. Moreover, the paper provides two (?) continuity equations (4 and 6) - which one should be used?
- The drag model employed also comprises a correction based on volume fraction, which is nonsensical when studying a single particle.
- The authors state that they use the k-epsilon turbulence model to conduct simulations that are all laminar - this is very puzzling.
- The transfer of data at processor boundaries is presented as an innovative feature, when in fact it is simply necessary if one aims to have a correct implementation of the model.
- Finally, the authors fail to identify and explain the reasons leading to the errors of the model. For instance, they observe an under-prediction of drag: this is easily explained by the very nature of the coupling between the Eulerian and Lagrangian frameworks. The momentum transferred from the particle to the fluid yields a non-zero velocity component in the negative vertical direction. When interpolated back to the particle to estimate drag, this results in the under-estimation of the relative velocity of the particle, and therefore the under-estimation of drag. As observed in the manuscript too, this effect is logically mitigated as the radius of the kernel is increased, since momentum is then spread over a larger region and therefore the local flow disturbance has a smaller magnitude. These observations and the dynamics of the coupling between the particle and the mesh have long been known in the community, and described in many previous papers, yet this is not explain is this manuscript.

For all these reasons, and due to the absence of any novel aspect, I must unfortunately recommend the rejection of the manuscript. If the authors plan on working further with filtered Euler-Lagrange modelling, I would kindly suggest to have a look at the recent work of S. Balachandar, University of Florida, J. Capecelatro at Michigan Uni, or J. Horwitz from Stanford Uni. My best wishes for your future research endeavours.

Author Response

The authors appreciate the reviewer’s favorable review and constructive suggestions. The suggestions were fully accommodated to improve the paper.

Attached please find the rebuttal.

Reviewer 2 Report

This manuscript has described a kernel-based averaged method for the fluid velocity used in the drag model. In addition, an expansion coefficient is used for the volume fraction in the drag model. Overall, an improved and smoother volume fraction field is obtained, as a result, the particle settling velocity is also smoother. The manuscript is well-written and organized. I only have minor comments and several questions for discussion, thus a minor revision is suggested.

  1. Equation (2), please double check the sign of the added mass coefficient.
  2. Equation (4) and (6) are two sets of continuity equation, however, the volume fraction field is directly obtained from DEM, so why do we need equation (6)? Please clarify.
  3. The author has compared the divided method with the proposed method. However, there are several other averaging methods in CFD-DEM, such as laplacian method. It will be helpful if the author can discuss the advantage of the proposed method over laplacian method as well.
  4. The authors argued that the added mass coefficient CA needs to be be increased for particle settling with low Reynolds number (the particle settling velocity is usually lower at the initial transition stage). However, added mass coefficient 1.5 or 2 seems not typical for spherical particles. Please justify these values. It seems that it is also possible that the drag model does not work well at low particle Reynolds number, maybe we can try other drag models that are well validated for low particle Reynolds numbers?
  5. It seems that the expansion coefficient Ep is introduced to adjust the averaged volume fraction used in the drag model, and it is shown that Ep=1.0 works the best, this corresponds to a larger fluid volume for the averaging. This way, the fluid volume fraction is closer to 1. Could you justify this adjustment? What if we directly set the fluid volume fraction to be 1 in the drag model? 
  6. Could the author discuss the applicability when particle flow is dense  (i.e., multiple particles)

Author Response

(The authors gave the same response as above.)

Reviewer 3 Report

The detailed suggestions are in the attached file.

Author Response

(The authors gave the same response as above.)

Round 2

Reviewer 3 Report

Dear authors,

The manuscript has been improved, but it needs to be polished again in terms of English language.

Please find attached the manuscript with the comments implemented there too.

Best regards
